# A Federated Learning and Deep Reinforcement Learning-Based Method with Two Types of Agents for Computation Offload

**DOI:** 10.3390/s23042243

**Published:** 2023-02-16

**Authors:** Song Liu, Shiyuan Yang, Hanze Zhang, Weiguo Wu

**Affiliations:** School of Computer Science and Technology, Xi’an Jiaotong University, Xi’an 710049, China

**Keywords:** mobile edge computing, computation offloading strategy, multi-agent system, deep reinforcement learning, federated learning

## Abstract

With the rise of latency-sensitive and computationally intensive applications in mobile edge computing (MEC) environments, the computation offloading strategy has been widely studied to meet the low-latency demands of these applications. However, the uncertainty of various tasks and the time-varying conditions of wireless networks make it difficult for mobile devices to make efficient decisions. The existing methods also face the problems of long-delay decisions and user data privacy disclosures. In this paper, we present the FDRT, a federated learning and deep reinforcement learning-based method with two types of agents for computation offload, to minimize the system latency. FDRT uses a multi-agent collaborative computation offloading strategy, namely, DRT. DRT divides the offloading decision into whether to compute tasks locally and whether to offload tasks to MEC servers. The designed DDQN agent considers the task information, its own resources, and the network status conditions of mobile devices, and the designed D3QN agent considers these conditions of all MEC servers in the collaborative cloud-side end MEC system; both jointly learn the optimal decision. FDRT also applies federated learning to reduce communication overhead and optimize the model training of DRT by designing a new parameter aggregation method, while protecting user data privacy. The simulation results showed that DRT effectively reduced the average task execution delay by up to 50% compared with several baselines and state-of-the-art offloading strategies. FRDT also accelerates the convergence rate of multi-agent training and reduces the training time of DRT by 61.7%.

## 1. Introduction

In recent years, with the development of mobile smart devices and 5G, many computationally intensive applications with low latency requirements have emerged, such as autonomous driving [1], virtual reality and augmented reality [2], online interactive gaming [3], and video streaming analysis [4]. These applications all have high requirements for quality of service (QoS). However, mobile devices (MDs) have limited computing power and are challenged by the growing demands for application computing power and increasingly stringent latency requirements.

To overcome this challenge, the mobile edge computing (MEC) [5] paradigm, as the core technology of 5G, pushes the computing resources on the network edge that is much closer to the MDs, thus relieving the network congestion and task delay of traditional centralized cloud computing. Unlike traditional cloud servers, MEC servers are not very resource-rich. Therefore, MDs need an efficient computation offloading strategy to determine whether to offload the tasks generated by MDs to MEC servers or the cloud server for execution, so as to fully utilize the computational resources to meet the quality of experience (QoE) of MDs and minimize the task execution latency. However, the uncertainty of computing tasks and the time-varying nature of wireless channels make it difficult for accurate and appropriate computation offloading decisions. 

Reinforcement learning (RL) is a method for learning “what to do (i.e., how to map the current environment into an action) to maximize the numerical revenue signal” [6]. With the rise of artificial intelligence, the deep reinforcement learning (DRL) that combines RL and deep learning is considered as an effective method to find asymptotically optimal solutions in time-varying edge environments [7]. Without any prior knowledge, DRL can capture the hidden dynamics of the environment well by enhancing the intelligence of the edge network, so as to learn strategies and achieve optimal long-term goals through context-specific repeated interactions. Such a property allows DRL to show its unique potential in designing computation offloading strategies in dynamic MEC systems. In the DRL method, the user data are transmitted to a central server for model training, and the agent deployed on MDs learns the strategy. The centralized DRL will not only put pressure on the wireless network but also lead to the risk of user data leakage. However, the existing DRL-based methods [8,9,10] rarely consider the issue of data privacy protection. 

To address these problems, this paper proposes a federated learning (FL) and DRL-based method with two types of agents for computation offload, named FDRT. FL [11] decouples the model from the user data and aggregates the model according to the uploaded local model parameters, thus achieving the balance between data privacy protection and data sharing. Applying FL to the MEC environment can realize decentralized distributed training and accelerate the parameter transmission and training speed of DRL agents. FDRT uses two types of DRL agents to better explore other MEC server resources for making the optimal offloading decision, thus achieving the goal of minimizing the task execution delay of each MD. The major contributions of this paper are summarized as follows. 

We proposed a multi-agent collaborative computation offloading strategy, named DRT. By building a collaborative cloud-side end MEC system, DRT was used to design a double deep Q-network (DDQN)-based mobile device agent and a dueling DDQN (D3QN)-based MEC server agent, which correspondingly decided to compute tasks locally on MD or offload tasks to MEC server. DRT enabled the offloading strategy to consider task information, network status, and nearby MEC server resources to ensure the optimal decision for minimizing the execution delay of tasks.We proposed an FL-based multi-agent training method, FDRT, to optimize the training of DRT. In FDRT, the MEC server aggregated the network parameters of MDs within its coverage to obtain the semi-global model, and the global model was aggregated from the semi-global models of nearby MEC servers, which reduced the parameter transmission of traditional FL training and the network overhead and thus improved the system’s QoS. Meanwhile, FDRT enabled the privacy protection of user data.We conducted two sets of simulation experiments. The experimental results showed that the proposed DRT achieved significant performance improvements over several baseline and state-of-the-art computation offloading strategies, reducing the average task execution delay by up to over 50%. We also demonstrated the effectiveness of FDRT, which reduced the training time of DRT by 61.7% and the average task execution delay by 2.8%.

The rest of the paper is organized as follows. Section 2 introduces the related work. Section 3 gives a detailed description of the system model, followed by the methodology in Section 4. In Section 5, we describe the proposed training optimization method. Section 6 presents the extensive experiments and result analysis. Finally, we conclude the paper in Section 7.

## 2. Related Work

The mathematical optimization method is a type of classical solution with a long history of computation offloading strategies, such as mixed integer programming (MIP), game theory, and heuristic search. Recently, Hazra et al. [12] propose a heuristic-based transmission scheduling strategy and a graph-based task offloading strategy using MIP in a high-traffic scenario for industrial IoT applications with a collaborative fog and cloud environment. Chen et al. [13] propose a non-cooperative game algorithm to find the Nash equilibrium solution with the optimization object of energy consumption and time delay for computation offloading of MEC applications. These methods are suitable for solving static optimization problems but not enough for dynamic optimization problems or sequence optimization problems. When a system disturbance occurs, mathematical methods need to reschedule or reoptimize. Moreover, when there are many user devices, these methods have a high computational complexity. For example, the time complexities of [12] and [13] are *O*(*N*^3^) and *O*(2*^N^*), respectively, where *N* is the number of user devices in the network. 

In recent years, learning-based approaches have become the dominant solutions to computation offloading strategies for MEC environments. Learning-based approaches mainly include multidisciplinary techniques, such as collective intelligence, constraint satisfaction problem (CSP), and machine learning, which have been widely used in different computer systems and network scenarios.

The properties of collective intelligence make it a practical model for algorithms to solve complex problems. The most commonly used algorithms are particle swarm optimization (PSO) algorithms [14,15]. Hussein et al. [16] efficiently distribute tasks to edge nodes and improve the response time of IoT applications by leveraging the global search capability of the ant colony algorithm. Rodrigues et al. [17] use the PSO to reduce the transmission and processing latency for minimizing the total end-to-end latency of partially offloaded tasks. Yadav et al. [18] combine a genetic algorithm and a particle swarm algorithm and propose a hybrid model to achieve near-optimal computation offloading for IoT applications while minimizing the total completion time and energy consumption.

The computation offloading problem can be defined as CSP, which has constraints such as QoS, heterogeneity of MDs, and dynamics of task generation. Wang et al. [19] formulate the CSP of computation offloading by combining user mobility, task properties, and network constraints to reduce the task execution delay in the MEC infrastructure. Kamal et al. [20] use the CSP formulation in conjunction with a minimal conflict scheduling algorithm to achieve a balanced load on MEC server resources with minimal energy consumption.

Due to the strong generalization ability, the machine learning models are also widely used for the task offloading strategy. Rahbari et al. [21] use a classification regression tree to select the most suitable edge device for offloading, and they also consider parameters such as authentication, integrity, availability, capacity, speed, and cost to minimize the time and energy consumption. Bashir et al. [22] use logistic regression to calculate the load of each edge node and propose a dynamic resource allocation policy. Ullah et al. [23] use K-means clustering approach to provide efficient task scheduling based on resource requirements in terms of CPU, I/O, and communication, thus improving the utilization of edge devices. To determine the combination of different devices and dynamic tasks, Rani et al. [24] propose a deep learning model to address the speed, power, and security challenges, while meeting the QoS.

RL is often combined with deep learning in order to generalize previously unseen data in terms of the environment, state, and behavior. In the MEC environment, Huang et al. [25] implement a DRL model that determines whether the task is computed locally or entirely handled by the MEC server. Zhou et al. [26] design an agent based on the double deep Q-network (DDQN) approach and deploy the agent on the MEC server to perform computation offloading decisions for all users simultaneously. In the context of vehicular networking, Zhao et al. [27] propose an algorithm based on duel deep Q-learning (DDQL) to implement distributed agent decision making, increasing the scalability and flexibility of the practical implementation of the algorithm. Yun et al. [8] use a genetic algorithm for user devices to determine the optimal offloading decision and propose a multi-actor deep Q-network agent for the MEC server to achieve dynamic resource allocation. Zhou et al. [9] formulate the optimization problem of computation offloading, service caching, and resource allocation as mixed-integer non-linear programming (MINLP) and use the Deep Deterministic Policy Gradient (DDPG) algorithm to find the optimal strategy to minimize the long-term energy consumption. Zhang et al. [10] use the reconfigurable intelligent surface (RIS) technique to adjust the phase shift and amplitude of reflective elements for improving the wireless network link status and energy efficiency and also use the DDPG algorithm for the computation offloading strategy. Li et al. [28] also design a offloading strategy based on a DDPG approach where all computing tasks sequentially decide the computation location through an agent. Koo et al. [29] adopt a Q-learning algorithm to find a task offloading strategy through the device-to-device communication in the MEC environment. They cluster nearby devices and use the head device of each cluster making the decision to reduce the computing complexity of agents. Under the conditions of power constraint of IoT devices and wireless charging, Wei et al. [30] propose agents based on post-state learning algorithms, which are deployed on MDs for decision making. In cyber-twin networks, Hou et al. [31] achieve fast task processing, dynamic real-time task allocation, and low training overhead by using a multi-agent deep deterministic policy gradient approach. 

There are several main drawbacks of the machine-learning-based methods. First, most of the existing methods use single-type DRL agents to accomplish the computation offloading decisions. They mainly consider the resources of MDs, connected MEC servers, and cloud servers but ignore the resources of other MEC servers in the MEC environment. In addition, it is difficult for MDs to access the resource information of MEC servers and cloud server, and accessing this information also causes decision latency. Second, in traditional machine learning architectures for MEC environment, data producers must frequently send and share data with third parties (e.g., MEC servers or cloud servers) to train their models. This not only causes the risk of user data privacy disclosure but also has a high demand for network bandwidth. In the time-varying wireless network environment, high communication overhead will undoubtedly have a negative impact on the decision delay.

To address these problems, we design two types of agents, a DDQN-based mobile device agent and a D3QN-based MEC server agent, to make the optimal offloading decision by considering the resources of near-neighbor MEC servers jointly. We apply the FL to the multi-agent collaborative training. FL can effectively protect user data privacy by keeping the data localized while only transmitting local model parameters for global model training. It has been applied in multiple domains in the MEC environment, including privacy protection and communication optimization for large-scale model training [32,33], content caching [34], malware and anomaly detection [35], task scheduling and resource allocation [36,37], computation offloading [38], etc. However, in most of the MEC federated learning systems, the parameters of all MDs are transmitted to an MEC server or the central cloud, which will not only cause a high network overhead and occupy the network resources in the core network or side layer but will also increase the latency of users waiting for the global model parameters, therefore reducing the system’s QoS. We also optimize the FL training efficiency by building a decentralized federation learning system model. Each MEC server and MD deploys an MEC server agent and a mobile device agent, respectively. Each MEC server additionally deploys a mobile device agent model for the semi-global model of all mobile devices within its current base station range. The MEC server aggregates the semi-global model from the nearby MEC servers to generate the global model, which reduces the parameter transmission of traditional FL training and the network overhead, especially compared with a central cloud-based federation learning model. Table 1 summarizes the comparison between our work and recent representative related work. For the complexity, *N* denotes the number of devices in the network, *T* denotes the number of episodes for model training, and *H* denotes the number of steps per episode. 

## 3. System Model

### 3.1. Network Model

In this paper, we considered the MEC system model with cloud-side end collaboration as shown in Figure 1. In the model, the scenario was seamlessly covered by *M* BSs that provided computation offloading services to *N* user MDs distributed within their range through a 5G wireless communication network, where the BSs were connected to each other via optical fiber. Each BS was equipped with an MEC server to provide computing power, so that the BS could compute a variety of tasks to meet the user’s needs. The user’s MD generated computing tasks that could be computed on the device or offloaded to an MEC server for execution. In addition, the BSs were connected to the core network via high-speed optical fiber, which in turn exchanged data with the central cloud, so that the MD could also offload tasks to the central cloud through the BS. The MDs were denoted as a set $U=\left\{u_{1},u_{2},\ldots ,u_{n},\ldots ,u_{N}\right\}$, and the BSs were denoted as a set $E=\left\{e_{1},e_{2},\ldots e_{m},\ldots ,e_{M}\right\}$. 

To facilitate the subsequent modeling, a time slot model was used to discretize the time into equal time intervals. The length of the time slot was denoted as $T_{len}$, and the index was denoted as$\;t=0,1,2,\ldots ,\;T_{len}$. The size of $T_{len}$ was set as the coherence time. The channel could be kept constant during the coherence time. In communication systems, the communication channel may change with time. This channel variation is more significant in wireless communication systems due to the Doppler effect. Table 2 presents the main notations that are used in this paper. 

### 3.2. Communication Model

In the MEC system network with cloud–edge–end collaboration, two main communication methods were included, i.e., edge-to-end wireless communication and edge-to-edge and edge-to-cloud wired communication. The transmission rate between BSs $e_{m}$ and $e_{m^{\prime }}$ was denoted as $v_{m,m^{\prime }}^{E}$, and the transmission rate between the BS $e_{m}$ and the central cloud was denoted as $v_{m}^{E,C}.$ The two kinds of communications both obeyed stable and independent random processes with probability distribution functions as $fun_{E}\left(v^{E}\right)$ and $fun_{E,C}\left(v^{E,C}\right)$, respectively. Next, the wireless communication methods between MDs and BSs are described and modeled in detail.

In order to minimize the execution delay of the computing tasks for MDs, we constructed a wireless communication model based on 5G technology. MDs may need to compute offload every *T* time to transmit data to the BS; therefore, the orthogonal frequency division multiplexing (OFDM) technique was used to assign different sub-channels to different devices to reduce the mutual interference between sub-channels and ensure the device transmission requirements.

In this paper, a Rayleigh fading model was constructed based on the free-space path loss model to simulate the channel in a dense building scenario that is very common in cities. In this scenario, there was no direct path between the transmitter and receiver, and the signal was attenuated, reflected, refracted, and diffracted by buildings or other objects. The channel remained stable within a time slot but changed from time slot to time slot. The channel gain $h_{n,m}\left(t\right)$ of the wireless channel is calculated by Equation (1) according to [26],
(1)$$h_{n,m}\left(t\right)=\beta_{n,m}\left(t\right)A_{d}{\left(\frac{c_{0}}{4\pi f_{c}d_{n,m}\left(t\right)}\right)}^{d_{e}},\;$$
where $\beta_{n,m}\;\left(t\right)$ denotes the channel fading factor of the Rayleigh distribution between $u_{n}$ and $e_{m}$ in the *t*th time slot, and its probability distribution function is$\;fun{\;}_{B}\left(\beta \right)$, $A_{d}$ denotes the radar gain of the BS connected to $e_{m}$, $c_{0}$ denotes the light speed in vacuum, $f_{c}$ denotes the carrier frequency of the BS connected to $e_{m}$, $d_{n,m}\left(t\right)$ denotes the distance between $u_{n}$ and $e_{m}$ in the *t*th time slot, and $d_{e}$ denotes the path loss index. 

Since the distance between a BS and its deployed MEC server was close, the transmission delay was negligible. According to the Shannon equation $C=B\;\log_{2}\left(1+S/N\right)$, combined with Equation (1), the transmission rate $v_{n,m}\left(t\right)$ between $u_{n}$ and $e_{m}$ in the *t*th time slot can be calculated by Equation (2),
(2)$$v_{n,m}\left(t\right)=B\;\log_{2}\left(1+\frac{p_{0}h_{n,m}\left(t\right)}{N_{0}}\right),$$
where B denotes the channel bandwidth between $u_{n}$ and $e_{m}$, $p_{0}$ denotes the transmission power of $u_{n}$, and $N_{0}$ denotes the Gaussian white noise power. Since the OFDM subchannels did not interfere with each other, the noise in the channel was only Gaussian white noise.

### 3.3. Task Model

At the beginning of the tth time slot, *N* MDs generate *N* indivisible tasks at the same time, and these tasks are independent. If no data were generated at the beginning of the tth time slot, $s_{n}\left(t\right)$ = 0. Since the data were actually generated in the $\left(t-1\right)$th time slot of $u_{n}$, which was processed from the *t*th time slot, they could be considered to be generated at the beginning of the tth time slot for the convenience of model representation. The computing task generated by $u_{n}$ at the beginning of the tth time slot was denoted by $d_{n}\left(t\right)∶=\left\{s_{n}\left(t\right),c_{n}\left(t\right)\right\}$.

### 3.4. Computation Model

The computing power of the central cloud is thousands of times greater than that of MDs and MEC servers. Therefore, the CPU frequency of the central cloud was set to infinity, and the CPU frequencies of the MD and MEC servers were set to fixed values, denoted as $f_{U}$ and $f_{E}$, respectively, where $f_{E}\gg f_{U}$. The computing power of the MD and MEC servers was relatively limited. If a task was computed on an MD or an MEC server, there may have been other tasks running, and the task could not be executed immediately. Therefore, we set task queues on both MD and MEC servers for the queuing tasks, where the “first in, first out” principle was applied to the queues.

The computing tasks of an MD could be executed locally on the device, offloaded to the MEC server of its BS or the MEC server of its neighboring BS, or offloaded to the central cloud for execution. Different computing modes led to different latencies, and the four computing modes were analyzed as follows.

(1)Local Computing. In the *t*th time slot, $u_{n}$’s task queue is denoted as $q_{n}^{U}\left(t\right)∶=\left\{b_{n}^{U}\left(t\right),c_{n}^{U}\left(t\right)\right\}$. When *t* = 0, $q_{n}^{U}\left(0\right)=\left\{0,0\right\}$, i.e., there is no task in the queue. If the computing task $d_{n}\left(t\right)$ of $u_{n}$ is executed locally, the task will be added to the local queue, and its execution delay $l_{n}^{L}\left(t\right)$ is calculated by Equation (3) according to [26].
(3)$$l_{n}^{L}\left(t\right)=\frac{s_{n}\left(t\right)c_{n}\left(t\right)+c_{n}^{U}\left(t\right)}{f_{U}}.$$

(2)MEC Computing. In the *t*th time slot, $e_{m}$’s task queue is denoted as $q_{m}^{E}\left(t\right)∶=\left\{b_{m}^{E}\left(t\right),c_{m}^{E}\left(t\right)\right\}$. If $d_{n}\left(t\right)$ is executed on $e_{m}$, the task will be added to the task queue of the MEC server, and its execution delay $l_{n}^{E}\left(t\right)$ is calculated by Equation (4) according to [26]. Since the resulting data of the task are usually very small, the transmission delay of the resulting data can be neglected.
(4)$$l_{n}^{E}\left(t\right)=\frac{s_{n}\left(t\right)}{v_{n,m}\left(t\right)}+\frac{s_{n}\left(t\right)c_{n}\left(t\right)+c_{m}^{E}\left(t\right)}{f_{E}}.\;$$

(3)Near MEC Computing. If $d_{n}\left(t\right)$ is executed on one of its neighboring BS $e_{m^{\prime }}$, the task will be added to the task queue of the MEC server $e_{m^{\prime }}$, and the task execution delay $l_{n}^{E^{\prime }}\left(t\right)$ can be calculated by Equation (5) according to [26].
(5)$$l_{n}^{E^{\prime }}\left(t\right)=\frac{s_{n}\left(t\right)}{v_{n,m}\left(t\right)}+\frac{s_{n}\left(t\right)}{v_{m,m^{\prime }}^{E}}+\frac{s_{n}\left(t\right)c_{n}\left(t\right)+c_{m^{\prime }}^{E}\left(t\right)}{f_{E}}.$$

(4)Cloud Computing. If $d_{n}\left(t\right)$ is executed in the central cloud, its execution time delay $l_{n}^{C}\left(t\right)$ can be calculated by Equation (6) according to [26]. Since the computing power of the central cloud is overwhelmingly strong, the computational time of the task in the central cloud can be ignored.
(6)$$l_{n}^{C}\left(t\right)=\frac{s_{n}\left(t\right)}{v_{n,m}\left(t\right)}+\frac{s_{n}\left(t\right)}{v_{m}^{E,C}}.$$

## 4. Methodology

### 4.1. Problem Statement

Based on the established system model, we first formalized the optimization problem of minimizing the long-term computational latency of tasks for the entire MEC system with limited resource constraints. The execution location of the computing task $d_{n}\left(t\right)$ in the tth time slot of $u_{n}$ is denoted as $a_{n}\left(t\right)$. When $a_{n}\left(t\right)=0$, it means that $d_{n}\left(t\right)$ is computed locally; when $a_{n}\left(t\right)=1$, it means that the task is executed on the MEC server of its connected BS; when $a_{n}\left(t\right)=2$, it means that the task is executed on the MEC server of its neighboring BS; and when $a_{n}\left(t\right)=3$, it means that the task is executed on the central cloud server. Jointly with Equations (3)–(6), the execution delay of the computing task $d_{n}\left(t\right)$ is calculated by Equation (7),
(7)$$l_{n}\left(t\right)=\&𝟙\left(a_{n}\left(t\right)=0\right)l_{n}^{L}\left(t\right)+𝟙\left(a_{n}\left(t\right)=1\right)l_{n}^{E}\left(t\right)+\&𝟙\left(a_{n}\left(t\right)=2\right)l_{n}^{E^{\prime }}\left(t\right)+𝟙\left(a_{n}\left(t\right)=3\right)l_{n}^{C}\left(t\right),\;$$
where $𝟙\left(\;\right)$ is the indicator function. When the condition in the function bracket is true, the value of the function is 1; otherwise, it is 0. Then, the optimization problem can be expressed as Equation (8),
(8)$$\begin{array}{c} P1:\underset{A\left(t\right)}{\min}\underset{\tau \to \infty }{\lim}\frac{1}{\tau }\overset{\tau -1}{\underset{t=0}{\sum }}\overset{N}{\underset{n=1}{\sum }}l_{n}\left(t\right)\\ s.t.\left\{\begin{array}{c} \;C1:a_{n}\left(t\right)\in \left\{0,\;1,\;2,\;3\right\}\\ C2:0\leq d_{n,m}\left(t\right)\leq D \end{array}\right. \end{array},\;$$
where $A\left(t\right)$ denotes the set of tasks’ execution locations of all MDs in the *t*th time slot, and D denotes the coverage radius of BS. Constraint C1 ensures that the task can only be executed once. Constraint C2 ensures that all user MDs are active within the coverage area of the BS and another BS continues to provide service when the MD exceeds the maximum service distance of the current BS. 

This optimization problem is an NP-hard problem. To effectively solve this problem, we proposed a DRL-based strategy with two types of agents for computation offload, i.e., DRT. DRT first divided the problem into two sub-problems, i.e., whether the MD executes computing tasks locally or which of the three computing modes of MEC computing, near-MEC computing, or cloud computing the task should perform. According to these two sub-problems, we approximately modeled them as a Markov decision process (MDP) and designed two types of DRL agents.

### 4.2. DDQN-Based Mobile Device Agent

We designed a DDQN-based mobile device agent, which was deployed on the MD to decide whether the MD executed computing tasks locally. The agent could make offloading decisions based only on the current computing task information, its task queues, and the network transmission rate between the MD and its connected MEC server. Since the agent was deployed on the MD, it could easily obtain such information. The three key elements of using the MDP to model the agent, including state space, action space, and reward function, are described as follows.

#### 4.2.1. State

Due to the limited computing power of MDs, it was not suitable to deploy a complex agent on the MD. We minimized the state space and defined the state $s_{n}^{U}\left(t\right)\in S^{U}$ of the mobile device agent in the tth time slot, as shown in Equation (9). The state space includes the size of computed data generated by a task, the number of CPU cycles required to process 1 bit of the data, the number of tasks in the local queue, the total number of CPU cycles to compute tasks in the local queue, and the transmission rate between the MD and its connected MEC server.
(9)$$s_{n}^{U}\left(t\right)∶=\left\{d_{n}\left(t\right),q_{n}^{U}\left(t\right),v_{n,m}\left(t\right)\right\}=\left\{s_{n}\left(t\right),c_{n}\left(t\right),b_{n}^{U}\left(t\right),c_{n}^{U}\left(t\right),v_{n,m}\left(t\right)\right\}.\;$$

#### 4.2.2. Action

The goal of the mobile device agent was to choose the optimal action to minimize the task execution delay based on the current state. The agent was responsible for deciding whether to execute the computing task locally or not at each time slot. The action of the agent is represented by $a_{n}^{U}\left(t\right)$, which is defined by Equation (10),
(10)$$a_{n}^{U}\left(t\right)\in A^{U}∶=\left\{0,1\right\},\;$$
where *A^U^* denotes the action set of the agent. When $a_{n}^{U}\left(t\right)=0$, it means the task is executed locally, and $a_{n}^{U}\left(t\right)=1$ means that the task is offloaded, and the subsequent decision is made by the agent of MEC server.

#### 4.2.3. Reward

In general, the reward function was related to the optimization goal, which was to minimize the long-term computation latency of all tasks in the entire MEC system. RL was an effective method to maximize numerical benefits, and we used the RL to construct the reward function. Therefore, the value of reward function was negatively correlated with the value of the optimization problem. The value of the reward function, $r^{U}\left(s_{n}^{U}\left(t\right),a_{n}^{U}\left(t\right)\right)\in R^{U}$, where *R^U^* is the reward space, for the MD agent to take action $a_{n}^{U}\left(t\right)$ under the state $s_{n}^{U}\left(t\right)$ in the tth time slot is calculated by Equation (11) according to [39],(11)$$\begin{array}{l} r^{U}\left(s_{n}^{U}\left(t\right),a_{n}^{U}\left(t\right)\right)=\left\{\begin{array}{l} -l_{n}^{L}\left(t\right),\;\;\;\;\;\;\;\;\;\;\;\;\;\;\;\;\;\;\;\;\;\;\;\;\;\;\;\;\;\;\;\;\;\;\;a_{n}^{U}\left(t\right)=0\\ -\left(\frac{s_{n}\left(t\right)}{v_{n,m}\left(t\right)}+time_{n,m}\left(t\right)\right),\;\;\;\;a_{n}^{U}\left(t\right)=1 \end{array}\right. \end{array},$$
where $time_{n,m}\left(t\right)$ indicates the subsequent execution time of the task $d_{n}\left(t\right)$ when it is offloaded for execution, and it is described in Section 4.3.3. 

#### 4.2.4. DDQN Model Training

We used the DDQN model [39] to train the mobile device agent to obtain the optimal offloading decision, which could reduce the high computational complexity caused by the state space explosion in the MDP. The DDQN model adopted an experience playback method to decouple the action selection and *Q* value calculation, where *Q* is the value of the reward function to take an action. During the training process, the agent maintained an experience playback pool and saved the transfer quads (*s*, *a*, *r*, *s*’) of each time slot in the experience playback pool. The elements of the transfer quad were the current state, the action, the reward value taking an action under current state, and the next state taking an action, respectively. The agent randomly selected a small batch of samples in the experience playback pool to update the parameters of the network, that is, randomly selected some previous experience to learn. The update of the experience playback pool followed the principle of “first in, first out”. The agent maintained the current network *Q*(*s*, *a*; *θ*) and the target network $\hat{Q}$(*s*, *a*; $\hat{\theta }$), where *Q* is used to select actions, $\hat{Q}$ is used to evaluate the value of the selected action, and *θ* and $\hat{\theta }$ represent the parameters of *Q* and $\hat{Q}$. The target network regularly updated its own parameters using the parameters of the current network, that is, to copy the parameters of the current network. The experience playback method was beneficial to accelerate the training convergence.

Figure 2 shows the training process of the DDQN-based mobile device agent. In each single episode, we first initialized the system model according to the task information, the task queue information, and the wireless network transmission rate, including the parameter θ of the current network Q, the parameter $\hat{\theta }=\theta$ of the target network $\hat{Q}$, and the experience replay pool M. The model started iterative training when obtaining the initial state. In each time slot, the “ε-greedy” strategy was used to select actions. It obtained a random value within [0,1); if the value was greater than ε, i.e., a preset exploration rate, then it randomly selected an action to execute, and otherwise, it selected the action with the maximum output Q value of the current network. Then, the model executed the action, obtained the corresponding reward and the next state, and stored the transfer information into M. The agent selected batch samples from M to calculate the gradient of loss function and used the gradient descent method to back propagate the gradient to minimize the loss function. Finally, the current network converged to the optimal action-value function through continuous iterative training of the above steps.

The DDQN model could ensure that the agent still had the ability to explore other actions after the network roughly converged, so as to prevent it from falling into the local optimum. 

### 4.3. D3QN-Based MEC Server Agent

The dueling DDQN (D3QN) model [40] has advantages in generalizing learning across actions without imposing any change to the underlying algorithm. This feature of D3QN is very suitable for the MEC server agent with a large action space. Therefore, we designed the D3QN-based MEC server agent, which is deployed on the MEC server, to make decisions for task offloading, i.e., it decided to compute the task on the MEC server or offload the task to a neighboring MEC server or to the central cloud for execution. Similar to the mobile device agent, we used the MDP to model the MEC server agent. 

#### 4.3.1. State

The state $s_{n,m}^{E}\left(t\right)\in S^{E}$ of the MEC server agent $e_{m}$ at the arrival of the computing task from $u_{n}$ is defined as Equation (12), where $\tilde{M}$ indicates the number of the MEC server near $e_{m}$. The agent made offloading decisions based on the offloaded tasks from MDs, the task queue status of its own MEC server, and the task queue status of its neighboring MEC servers. Since the server agent was deployed on the MEC server of BS, and the BSs of telecom operators were also mutually trusted, the agent could easily access the task queue status of its neighboring MEC servers.
(12)$$\begin{array}{c} s_{n,m}^{E}\left(t\right)∶=\left\{d_{n,m}\left(t\right),q_{n,m}^{E}\left(t\right),q_{n,m}^{E_{1}}\left(t\right),..,q_{n,m}^{E_{\tilde{M}}}\left(t\right)\right\} \end{array}.\;$$

#### 4.3.2. Action

The goal of the MEC server agent was to decide the execution location of the offloaded task, i.e., choose the optimal action to minimize the task execution delay based on the current state. The MEC server agent was responsible for deciding the computing mode of the task. The action for task $d_{n,m}\left(t\right)$ is defined by Equation (13), where *A^E^* denotes the action set of the server agent. When $a_{n,m}^{E}\left(t\right)=0$, it means the task is executed on the local MEC server, $a_{n,m}^{E}\left(t\right)=1$ means that the task is offloaded to the central cloud for execution, and $a_{n,m}^{E}\left(t\right)=\left\{2,\ldots ,\tilde{M}+1\right\}$ means that the task is offloaded to a neighboring MEC server for execution, and the numerical value represents the number of neighboring servers.
(13)$$a_{n,m}^{E}\left(t\right)\in A^{E}∶=\left\{0,1,2,\ldots ,\tilde{M}+1\right\}.\;$$

#### 4.3.3. Reward

Similar to the MD agent, the value of reward function, $r^{E}\left(s_{n,m}^{E}\left(t\right),a_{n,m}^{E}\left(t\right)\right)\in R^{E}$, for the server agent taking action $a_{n,m}^{E}\left(t\right)$ under state $s_{n,m}^{E}\left(t\right)$ in the *t*th time slot, is calculated by Equation (14) according to [40]. The subsequent execution time of the task, $time_{n,m}\left(t\right)$, is given by Equation (15) according to [40] for different cases, i.e., when it is executed on a local MEC server ($a_{n,m}^{E}\left(t\right)=0$), offloaded to the central cloud for execution ($a_{n,m}^{E}\left(t\right)=$1), or offloaded to a neighboring MEC server for execution ($a_{n,m}^{E}\left(t\right)=\left\{2,\ldots ,\tilde{M}+1\right\}$). It is related to the task information, the task queue information, and the wireless network transmission rate.
(14)$$r^{E}\left(s_{n,m}^{E}\left(t\right),a_{n,m}^{E}\left(t\right)\right)=-time_{n,m}\left(t\right).\;$$
(15)$$time_{n,m}\left(t\right)=\left\{\begin{array}{c} \frac{s_{n,m}\left(t\right)c_{n,m}\left(t\right)+c_{m}^{E}\left(t\right)}{f_{E}},\;a_{n,m}^{E}\left(t\right)=0\\ \frac{s_{n,m}\left(t\right)}{v_{m}^{E,C}},\;a_{n,m}^{E}\left(t\right)=1\\ \frac{s_{n,m}\left(t\right)}{v_{m,m^{\prime }}^{E}}+\frac{s_{n,m}\left(t\right)c_{n,m}\left(t\right)+c_{m^{\prime }}^{E}\left(t\right)}{f_{E}},\;a_{n,m}^{E}\left(t\right)=\left\{2,\ldots ,\tilde{M}+1\right\} \end{array}\right..$$

#### 4.3.4. D3QN Model Training

The MEC server agent had larger and more complex state space and action space than the MD agent, especially the action space. When the action space was very large, D3QN performed better than the traditional DRL networks [40]. Therefore, we used the D3QN model to train the MEC server agent, so as to avoid the slow convergence of the MEC server agent and the inability to make timely decisions on computing task offloading. The training process of the D3QN model was similar to that of the DDQN model. The current network and the target network of D3QN model had the same functions as those of the DDQN model, but their structures were slightly different. The current network and the target network of the D3QN model had two sets of output parameters, which aggregated to output the Q value of each action. Due to the paper limitation, we will not describe these specifically, and the relevant network structure can be found in [40]. 

### 4.4. DRT

Based on these two types of agents and Equation (8), DRT obtained the optimal computation offload strategy for the MEC system through multi-agent cooperative training. The training process of DRT with multiple episodes is described as follow. 

First, the system model and the parameters of each agent network were initialized before multi-episode iterative training. In a single episode, each mobile device agent obtained the current initial state and started training from that state. If the mobile device agent offloaded computing tasks to its own connected MEC server, the MEC server agent started a single episode training. Finally, through continuous iterative training and learning over multiple episodes, the current network convergence of each mobile device agent and each MEC server agent approximated the optimal action-value function, that is, the optimal unloading strategy for all tasks in the entire MEC system was learned, which minimized the calculation delay of all tasks in the entire MEC system.

## 5. Optimization Method of Multi-Agent Training

### 5.1. Deployment of Two Types of Agents

Although DRT could dynamically and efficiently find the optimal computation offloading strategy, it also required a lot of computing resources to train agents. Therefore, the deployment of the two types of agents should be carefully considered in order to effectively use the computing resources in the MEC systems.

As described in Section 4, the MEC server agent is suitable to be deployed on the MEC server. For the mobile device agent, training it using the traditional RL training method on the MD would introduce two shortcomings. First, additional energy will be wasted by training separate agents for each MD. Second, MDs have limited computing power, and the cost of training the agent from scratch is too high. There are also several drawbacks if the mobile device agent is trained on the MEC server that the MD is connected to. (a) There will be communication overhead between the MD and the MEC server, which causes delays in making offloading decisions and makes it difficult to make decisions in real time. (b) If only the connected MEC server maintains the mobile device agent, it involves the migration of agent data when users move to another BS’s coverage. If all MEC servers maintain agents for each MD, it will not only result in resource waste but also bring in extra synchronization of agent data. (c) The privacy of MDs may be compromised as the uploaded training data may be privacy-sensitive, especially in industrial information scenarios. (d) Although the training data can be transformed to protect privacy, the data received by the MEC server would lose some relevancies compared with the source data, making it difficult to optimize the offloading decision or even making a worse offloading decision. (e) A large amount of training data is always transmitted from MDs to the MEC server, which puts a heavy burden on the wireless channel of BSs.

Therefore, we deployed the DDQN-based agent on MDs and the D3QN-based agent on the MEC server and proposed the FDRT optimization method for multi-agent training.

### 5.2. FDRT

We applied the FL to achieve the optimization method of the multi-agent training for the proposed DRT, named FDRT. As a cooperative framework of machine learning, FL can train models without accessing users’ private data, and it can achieve decentralized distributed training with the mechanism of aggregation model. DRT needs to interact with the environment frequently for multi-agent collaborative training, which will generate a large number of network communications, increase the computing pressure of resource-constrained mobile devices, and thus increase the latency of computation offload decisions. When combining the FL with DRT, due to the decentralized topology of FL, the mobile device agent only needs to communicate with its connected MEC server. The MEC server aggregates the local model parameters of MDs within its coverage and transmits the aggregation results to them. This efficient decentralized training model can greatly reduce network overhead, improve users’ QoS, and reduce the training time of the multi-agent model. 

In the multi-agent trained federation learning system model, each MEC server and each mobile device deployed an MEC server agent and a mobile device agent, respectively. Each MEC server also deployed an additional mobile device agent, which was used as the global model of all MDs within the BS of the server. The mobile device agent can be regarded as the client of the FL model, and its connected MEC server can be regarded as the parameter server. For the mobile device agent, the state space is the initial data set of the client of the FL model. The target network of the mobile device agent used the discounted sum of its output $\hat{Q}$ value and the reward as the target value of the action in the current state and used the difference between the target value and the *Q* value of the current network to calculate the local loss function of the client, so as to update the current network parameters. The client updated the neural network parameters based on the local data set and uploaded the trained network parameters to the server. The server aggregates these updated parameters to obtain a global parameter and then transmitted it back to the clients for the next round of local training. Before parameter aggregation, each client performed multiple local training and parameter updates in a round of training. 

### 5.3. FDRT Training and Semi-Global Aggregation

In the FL model for multi-agent training, the workflow was basically the same for all BSs. The workflow for a single BS with multiple MDs scenario is shown in Figure 3. First, a global model of the mobile device agent was initialized on the MEC server. The initial model parameters may be different for different MEC servers. The current network parameters of the model are distributed to all MDs within the coverage area of the BS. The MD synchronized the received parameters to the local current network and the target network. Then, the mobile device agent started local training, and the parameters were transmitted to the MEC server to which the MD was connected after every *F* times training in a round. If the device moved to the coverage of another BS during training, it transmitted the parameters of the current network to the reconnected MEC server, rather than transmitting the parameters back to the previous connected MEC server. The MEC server received the network parameters of all MDs in its coverage area and averaged the parameters to obtain a new network parameter, which was referred to as a semi-global model parameter. Further, the MEC server obtained the semi-global model parameters from its neighboring MEC servers to generate the global model parameter based on the weighted average of the number of devices in the coverage area. If a neighboring MEC server did not have any MDs, the semi-global model parameters of this server were not aggregated. Finally, the MEC server transmitted the global model parameters back to MDs. The MD synchronized the new parameters to the local current and target networks and then started a new round of training.

In traditional FL, local model parameters of all clients are transferred to a server for global parameter aggregation. In the proposed FDRT, mobile devices transmitted local model parameters to their connected MEC servers to calculate a semi-global parameter, and the MEC server aggregated the semi-global parameters of their neighboring MEC servers to obtain the global parameter. Therefore, compared with traditional FL, FDRT can alleviate the pressure of wireless network communication, reduce the delay of global model parameter transmission, and improve the efficiency of model training. 

Algorithm 1 describes the training process of the mobile device agent of the FDRT. The training process of MEC server agent was similar. The time complexity of FDRT is *O*(*TH*), where *T* is the number of training episodes, and *H* is the number of steps per episode. The complexity of FDRT mainly came from the training process of the DDQN agent and the D3QN agent, which was consistent with the complexity of other DQN-based methods [41].
**Algorithm 1** FDRT’s Mobile Device Agent Training.**INPUT:** maximum number of training episodes MAX_EPISODE, maximum number of training steps per episode MAX_STEP, learning rate α, discount factor γ, exploration rate ε, experience replay pool M capacity CAPACITY, number of batch samples BATH_SIZE, current network Q, target network $\hat{Q}$, target network update frequency C, numbers of training in a round of federated learning F
**OUTPUT:** parameter $\theta_{t}$ of current network
// $random\left(a,b\right)$ is a function that generates random numbers in range $\left[\left.a,b\right)\right.$
// $randint\left(a,b\right)$ is a function that generates random integers in range $\left[\left.a,b\right)\right.$
// memory.isfull() indicates whether the experience replay pool M is full
// $\theta^{e}$ denotes the global model parameters of the MEC server
**Initialization:** Wireless communication model between MDs and BSs; task model and queue model of MD, parameter θ of current network Q$,\;\mathrm{parameter}\;\hat{\theta }=\theta$$\;\mathrm{of}\;\mathrm{target}\;\mathrm{network}\;\hat{Q}$, experience replay pool M.
1:  **for** episode=0:MAX_EPISODE **by** 1 **do**
2:    Get initial state $s_{0}$;
3:    **for** t=0:MAX_STEP **by** 1 **do**
4:      
$x\leftarrow random\left(0,1\right)$;
5:      **if**(x>ε) **then**
6:        
$a_{t}\leftarrow randint\left(0,2\right)$;
7:      **else**
8:        $a_{t}\leftarrow \underset{a}{\mathrm{argmax}}Q\left(s_{t},a;\theta_{t}\right)$;
9:      **end if**
10:      Perform action $\;a_{t}$ in the system model, get reward $r\left(s_{t},a_{t}\right)$ and next state $s_{t+1}$;
11:      $s_{t}\leftarrow s_{t+1}$;
12:      Put $I_{t}∶=\left(s_{t},a_{t},r\left(s_{t},a_{t}\right),s_{t+1}\right)$ in M;
13:      **if**(memory.isfull()) **then**
14:        **continue;**
15:      **end if**
16:      **if**$\left(\left(episode*MAX\_STEP+t\right)\;mod\;F==0\right)$
**then**
17:         Upload θt to connected MEC server;
18:         θt←θe;19:      **else**
20:        Randomly choose a batch sample from M to update parameter, θt←θt−α∇Lθt;
21:      **end if**
22:      **if**$\left(t\;mod\;C==0\right)$
**then**
23:        θ^t←θt;24:      **end if**
25:    **end for**
26:  **end for**
27:  **return** $\theta_{t}$;

Algorithm 2 describes the process of MEC server parameter aggregation of the FDRT. First, the MEC server obtained the parameters of all MDs in its coverage range and calculated the average value to generate the semi-global model parameters. Then, the MEC server obtained the semi-global model parameters of its neighboring MEC servers and calculated the global model parameter.
**Algorithm 2** FDRT’s MEC Server Parameter Aggregation.**INPUT:** mobile device agent parameters θu
**OUTPUT:** global model parameter θe
// u∈Ue means that MD u is within the coverage of MEC server e, where Ue denotes the set of MDs within the coverage of e
// e′∈Ee means that MEC server e′ is a server near e, where Ee denotes the set of servers near e
// get_MD_nume denotes the number of MDs within range of MEC server e
1:  θtempe←0;
2:  **for** u **in** Ue **do** 
3:      θtempe←θtempe+θu;4:  **end for**5:  θtempe←θtempe/Ue;6:  count←get_MD_nume;7:  θe←count*θtempe;8:  **for** e′ **in** Ee **do**
9:      **if**get_MD_nume′>0
**then**
10:          $\theta^{e}\leftarrow \theta^{e}+get\_MD\_num\left(e^{\prime }\right)*\theta_{temp}^{e^{\prime }}$;
11:          $count\leftarrow count+get\_MD\_num\left(e^{\prime }\right)$;
12:      **end if**
13:  **end for**
14:  $\theta^{e}\leftarrow \theta^{e}/count$;
15:  **return** $\theta^{e}$;

## 6. Experimental Results

### 6.1. Comparison Strategies

We conducted two sets of simulation experiments to evaluate the performance of the proposed DRT and the FDRT, respectively. To verify the effectiveness of DRT, we compared it with six baseline strategies and two state-of-the-art offloading strategies. To verify the training efficiency of FDRT, we compared it with DRT and a traditional federated learning method (TFL) [11]. As this work aimed to minimize the system latency, we mainly evaluated the reward value and the task execution delay under different strategies. The reward value could not only reflect the system latency but also reflect the convergence trend of the agent.

The baseline strategies are denoted as MDL, MDO, MDR, MSL, MSC, and MSR. MDL represents the strategy that all computing tasks are executed on MDs locally; MDO represents the strategy that all MDs offload their tasks to MEC servers to decide the specific execution location; MDR represents the strategy that all MDs randomly select tasks to compute locally or offload tasks; MSL represents the strategy that all MEC servers execute tasks locally; MSC represents the strategy that all MEC servers offload tasks to the central cloud; and MSR represents the strategy that all MEC servers randomly select tasks to compute locally, offload tasks to a neighboring MEC server, or offload tasks to the central cloud. 

In addition, two state-of-the-art computing task offloading strategies are the single MD agent (SMDA) method [27] and one MEC server agent (OMSA) method [26]. SMDA designs the DDQL-based agent deployed on MDs to make offload decisions based on the task information generated by current time slot, the task queue information of MD and its connected MEC server, and the network conditions of both. SMDA adopts the collaborative training of FL and blockchain to learn the optimal offloading strategy, which is consistent with the goal of our strategy, that is, to minimize the system latency. OMSA designs the DDQN-based agent deployed on the MEC server, which finds the offloading decision through DRL training to solve the MINLP optimization problem based on the task information, the task queue information of MDs and all MEC servers, and the network status of MDs and their connected MEC servers. The optimization objective of OMSA is to minimize the energy consumption with delay constraints. 

### 6.2. Simulation Setting

We simulated a collaborative cloud-side end MEC system, which consisted of multiple BSs and multiple MDs. Figure 4 shows the topology diagram of MEC servers. In the simulation system, the number of MEC servers was set to six. The shaded circles indicate that there are MDs within the range of the BSs, and the blank circles indicate that there are no MDs within the range of the BSs. The number of MDs in each BS was set to five. In the training process of agents, the maximum episodes and the maximum steps of each episode were set to 200. 

Table 3 shows the model parameters and their simulation values. We used the parameters of Huawei’s 5G AAU5619 wireless product (Huawei Technologies Co., LTD., Shenzhen, China) to simulate the BS transmitter and built a wireless communication model between MDs and BSs on this basis. In the MEC system, the wired communication model included the wired communication between BSs and between BSs and the central cloud. The computing model included the task model of MDs and the CPU frequency of MDs and MEC servers. Both the mobile device agent and the MEC server agent used the Adam optimizer and the mean square error loss function. The current network and target network structures of mobile device agents were four-layer fully connected neural networks, in which the numbers of hidden layer neurons were 128 and 64, respectively. The current network and target network structures of MEC server agent were a three-layer fully connected neural network and a one-layer two-branch network, in which the number of hidden layer neurons was 128. 

We designed a simple movement model to simulate the movement of MDs, which was used to generate the distance between BSs and MDs. The MD initially moved randomly in a clockwise or counterclockwise direction along the connection line between the MEC servers in Figure 4, and there was a probability of 0.00001 that the MD would turn around and move in the opposite direction during the movement. The moving speed of the MD obeyed a uniform distribution of *U*(29.6,30.4) m/s, which is the general walking speed of people.

The simulation experiments were performed on a server equipped with two Inter(R) Xeon(R) Gold 6248 processors at 2.50 GHz (Intel Corporation, Santa Clara, CA, USA). All experimental codes were implemented in Python, version 3.6.8, and the PyTorch library, version 1.8.2.

### 6.3. Performance of DRT

We used the reward values to evaluate the performances of different computation offloading strategies. Since the change trend of the task execution delay was exactly opposite to that of the reward, the larger the reward value was, the smaller the task execution delay was. 

Figure 5 shows the comparison results of the reward with the DRT, MDL, MDO, and MDR strategies during the training process. The ordinate represents the average value of the cumulative reward sum of all MDs per episode. As the number of training episodes increased, the rewards of all strategies tended to converge. DRT achieved the highest reward compared with the other baseline strategies, which meant DRT could learn the best offloading decision with minimal latency for MDs. In the first 50 episodes, the rewards of MDO and MDR increased gradually. This was because both MDO and MDR offloaded tasks to the MEC server for further decision making by the MEC server agent, i.e., the increasing trends reflected the training process of the D3QN-based MEC server agent. The MDO obtained a reward close to that of DRT. This was because when the wireless network was stable for most of the time, it was more conducive for the mobile device agent to offload tasks to the more powerful MEC server to reducing the execution delay of all tasks. Due to the limited computing power of MDs, local computing had a limited effect on reducing the overall task execution delay. For the same reason, the MDL received the lowest reward and the worst task execution delay.

Figure 6 shows the comparison results of the reward with DRT, MSL, MSC, and MSR strategies. The DRT still achieved the highest reward compared with the other baseline strategies, which meant DRT could fully utilize the computing resources of the MEC servers and the central cloud to make the best offloading decision. In the first 25 episodes, the rewards of both MSL, MSC, and MSR increased gradually. The reason was that the DDQN-based mobile device agent was learning the offloading strategy, which also reflected the convergence performance of the MD agent. The reward obtained by MSL was close to that of DRT. Although DRT considered the computing resources of neighboring MEC servers to offload tasks, these resources were also used by other MEC servers, so the overall improvement was slightly better than MSL. However, compared with MSC and MSR, DRT showed significant performance improvements.

Table 4 shows the comparison results of the average task delay between DRT and baseline strategies. The results of the single episode were selected from the multiple episodes before convergence, and the results of multiple episodes were calculated from the convergent episode. DRT reduced the average task execution delay by 41.2%, 5.9%, 17.2%, 2.8%, 31.7%, and 24.2% compared with MDL, MDO, MDR, MSL, MSC, and MSR, respectively. This was due to the fact that DRT adopted two types of agents to jointly learn the optimal offloading strategy, which could more comprehensively evaluate the entire MEC system resources to minimize the system latency.

Figure 7 shows the comparison results of the reward with DRT, SMDA, and OMSA during the training process. With the increase in the training episode, both SMDA and OMSA tended to converge, and DRT obtained the highest reward. Compared with SMDA and OMSA, DRT could fully utilize the computing resources of MDs, MEC servers, and the central cloud. The reason for the huge fluctuation of OMSA was that it filled the experience replay pool in the first episode, and the agent already started to learn and update the network parameters in this episode, which caused overfitting. Therefore, when OMSA encountered a large number of states different from previous ones, the learned strategies did not necessarily perform well. When the reward during 0-100 episodes was zoomed in on, it could be observed that SMDA was close to DRT, but DRT converged faster than SMDA and OMSA. This was because SMDA applied FL to achieve decentralized training, which effectively reduced the communication overhead. However, DRT divided the computation offload decision problem into two sub-decision problems, which were respectively solved by the two types of agents. In this way, the state and action space of DRT agent was much smaller than that of SMDA and OMSA. It could not only enable the agent to learn the optimal strategy faster but also reduce unnecessary data transmission between MDs and MEC servers. Since SMDA and OMSA used a single type of agent, they needed to obtain MEC server information or mobile device information, which increased the burden of wireless networks and led to higher decision delay.

Table 5 shows the comparison results of average task delay between DRT and SMDA and OMSA. DRT reduced the average task execution delay by 8.0% and 50.3% compared with SMDA and OSMA, respectively. When the FDRT optimization method was applied to DRT, the task execution delay was further reduced. The performance will be presented in Section 6.4. 

### 6.4. Performance of FDRT

Since FDRT combined the FL with DRT to optimize the training of multi-agent model, we compared FDRT with DRT and TFL. DRT used the original DRL method to train the model, and TFL [11] used a traditional federated learning method to train the mobile device agent.

Figure 8 shows the comparison results of the loss with FDRT, DRT, and TFL during the training process. The ordinate denotes the average values of the cumulative loss sum of all mobile device agents in each episode. The loss values were kept at 0 at first and then rose sharply. This was because the experience replay pools of mobile device agents were not full yet, so the agents had not started learning. In addition, FDRT and TFL accelerated the convergence of mobile device agents, and the losses of the three methods were finally stabilized at a certain value. It can be seen from the zoomed in sub-figure that DRT converged at about 60 episodes, while FDRT converged at about 23 episodes, saving 61.7% of the training time for the mobile device agents. 

Figure 9 shows the comparison results of the average task execution delay of FDRT, DRT, and TFL. It can be observed that the learned strategies of FDRT and TFL were basically the same as that of DRT. The FDRT sped up the training convergence of mobile device agents, while still maintaining the optimal learned strategy. In addition, FDRT aggregated fewer network parameters of mobile device agents in each MEC server and reduced the network transmission of parameters more than DRT and TFL. Compared with DRT, FDRT reduced the average task execution delay by 2.8%, indicating that the strategy learned by FDRT was better. This was because the semi-global aggregation parameter mechanism of FDRT enabled each mobile device agent to learn more state information, so that it was more likely to take the optimal action when the state of the MD changed. 

### 6.5. Discussion

The superior performance of this work over existing methods mainly comes from the following aspects. Compared with existing DRL-based methods with a single type of agent, DRT used two types of agents for MDs and MEC servers to collaboratively learn the offloading strategies. Hence, DRT could find the optimal decision by comprehensively considering all MEC server resources and network conditions in the MEC environment for minimizing the system latency. The DDQN agent and the D3QN agent also considered the application characteristics of MDs and MEC servers to reduce the high dimensionality of MDP. In addition, FDRT applied FL to DRT to accelerate the multi-agent collaborative training, while effectively providing user data privacy protection. The decentralized FL training method distributed the model aggregation among MEC servers to relieve the computing pressure of resource-constrained MDs. The semi-global aggregation mechanism reduced the parameter transmission and network overhead, which further shortened the delay of strategy decision. These optimizations were very effective for latency-sensitive and computationally intensive applications to quickly learn the computation offloading strategy in the time-varying MEC environment. Therefore, this work is very suitable for the real-word applications with a high requirement of task execution delay, such as autonomous driving, online interactive gaming, virtual reality applications, and video streaming analysis. 

Since this work focuses on the system latency, one of its main limitations is that it may not be able to fully meet the application scenarios with a high requirement of energy consumption. When combining energy consumption into our current model, it needs to consider the heterogeneity of different resources to formulate the multi-objective optimization problem. It needs more consideration to realize a balanced optimal offloading strategy among many objectives such as energy consumption, task delay, system utilization, etc. This will be one of our future work directions. In addition, this work is a binary computation offloading method. For many large-scale applications, the partial computation offloading strategy is more suitable. We will also study the partially offloading method in the future. 

## 7. Conclusions

In this paper, we proposed a computation offloading method based on DRL and FL with the goal of minimizing the task execution delay for a system of multi-base station multi-mobile device MEC networks with cloud-side end collaboration. First, we proposed a multi-agent computation offloading strategy using a DDQN MD agent and a D3QN MEC server agent, to collaboratively make decisions based on their own task information, resources, and time-varying network conditions. Second, we proposed a new FL-based parameter aggregation model that greatly reduced communication overhead and improved the training efficiency of the multi-agent DRL model, while avoiding user data privacy disclosure. Extensive simulation experiments validated the advantages of the proposed method in computing task execution delay and training efficiency over several baseline and state-of-the-art computation offloading methods. In future work, we will try to extend the scope of application of this work by jointly considering the energy consumption model to maximize the system utility with energy latency constraints. For large-scale real-world applications, it is also our future research direction to use the collaboration of DRL and FL, solving the partial computation offloading problem.

## Figures and Tables

**Figure 1 sensors-23-02243-f001:**
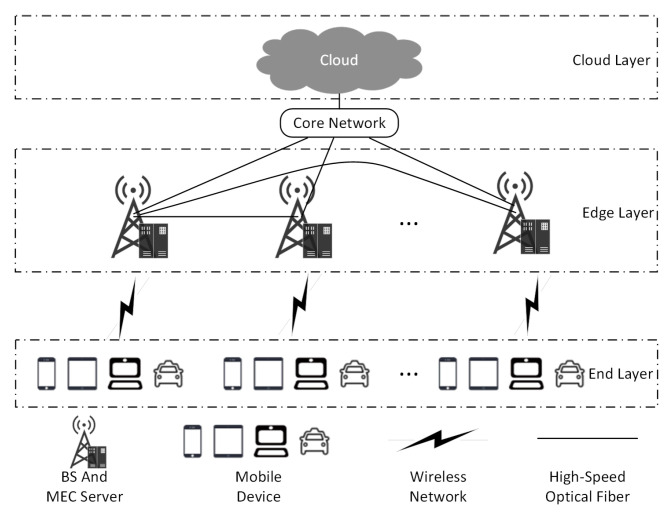
MEC system model of cloud–edge–end collaboration.

**Figure 2 sensors-23-02243-f002:**
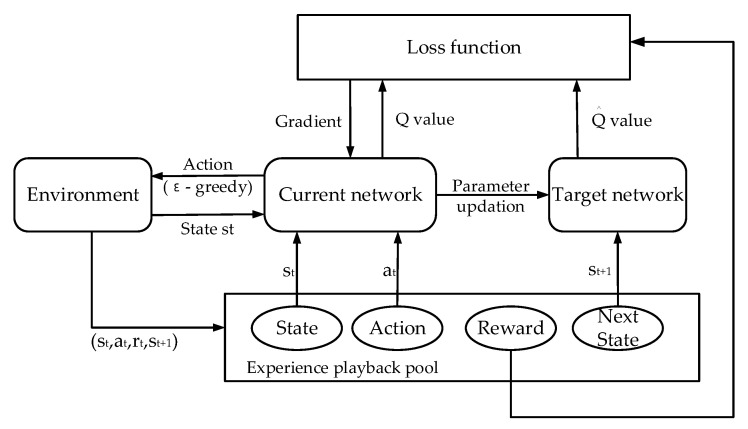
Training process of the DDQN-based mobile device agent.

**Figure 3 sensors-23-02243-f003:**
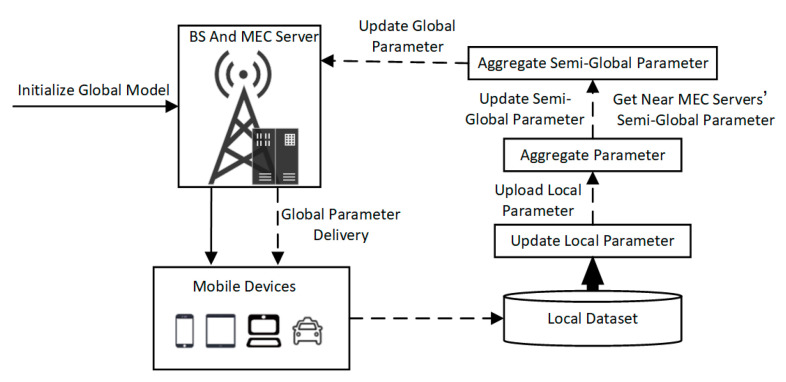
Federated learning system workflow for multi-agent training.

**Figure 4 sensors-23-02243-f004:**
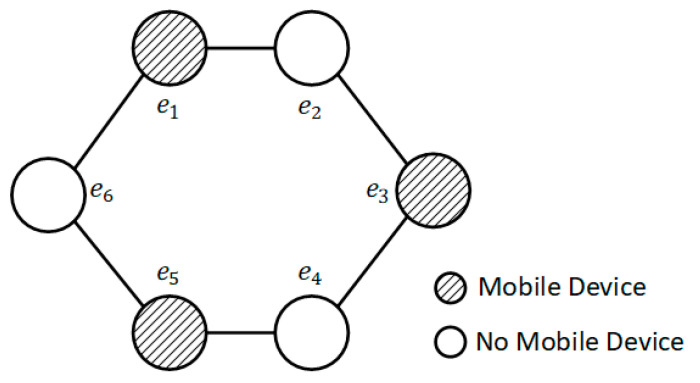
MEC server topology diagram.

**Figure 5 sensors-23-02243-f005:**
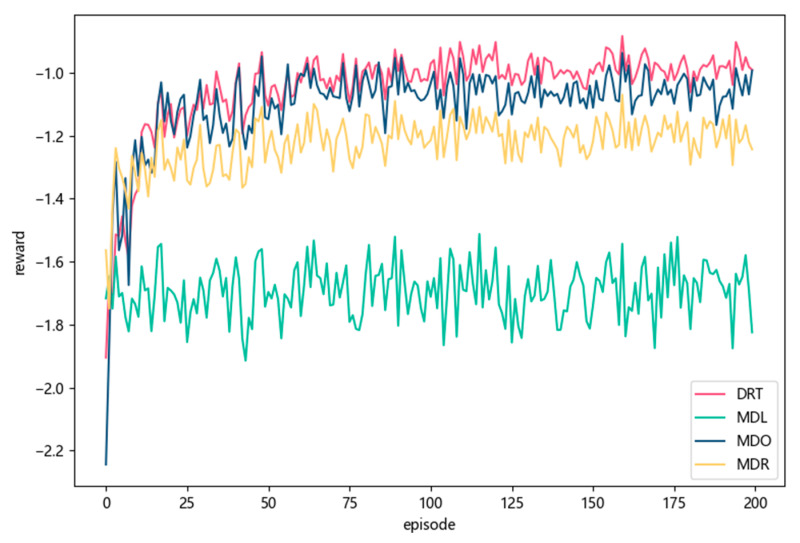
Comparison of reward with different strategies on MDs during training.

**Figure 6 sensors-23-02243-f006:**
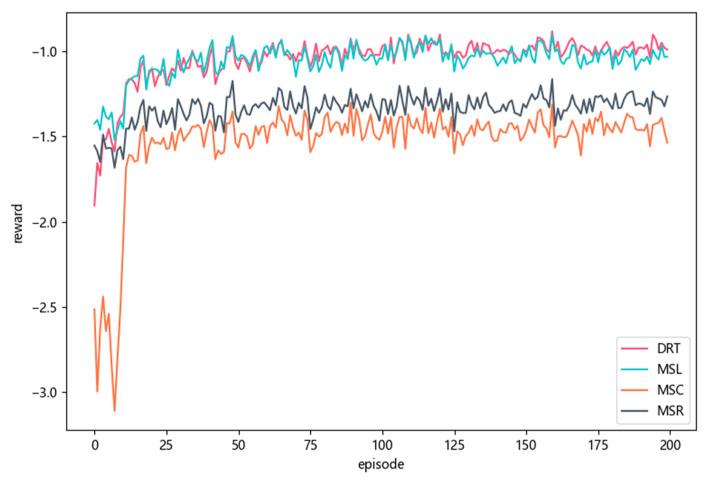
Comparison of reward with different strategies on MEC servers during training.

**Figure 7 sensors-23-02243-f007:**
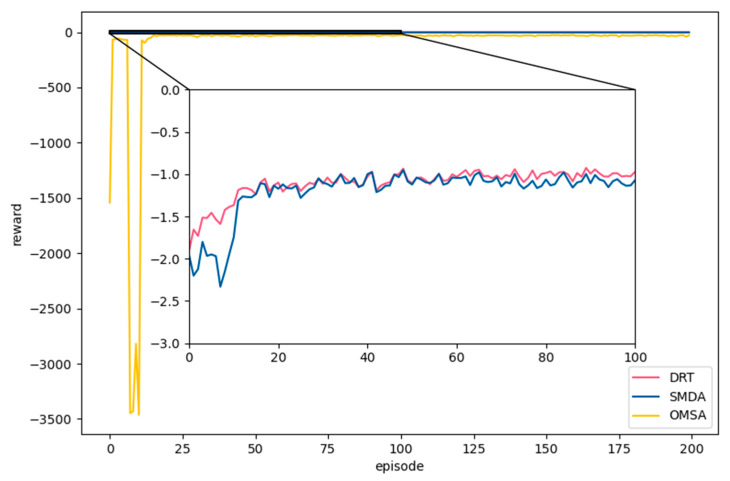
Comparison of reward with SMDA and OSMA.

**Figure 8 sensors-23-02243-f008:**
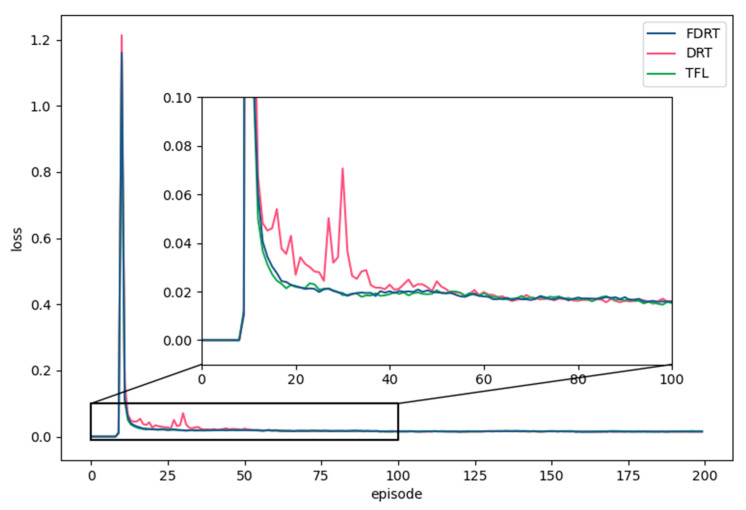
Comparison of losses of mobile device agents with FDRT, DRT, and TFL.

**Figure 9 sensors-23-02243-f009:**
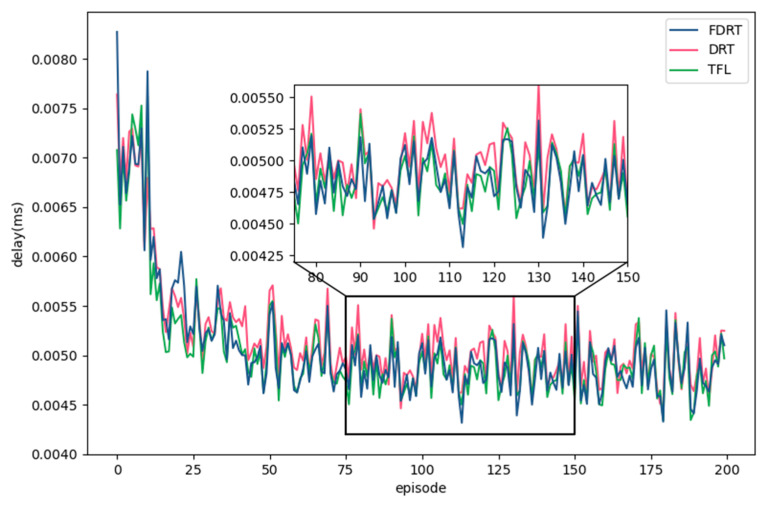
Comparison of average task execution delay of FDRT, DRT, and TFL.

**Table 1 sensors-23-02243-t001:** Comparison with existing works.

Existing Works	Main Method	Device Mobility	Data Privacy Protection	Network Training Optimization	Optimization Goal	Complexity
[12]	Heuristic, MIP	Static	No	—	Energy consumption and latency	*O*(*N*^3^)
[13]	Game theory	Dynamic	No	—	Energy consumption and latency	*O*(2*^N^*)
[8]	Genetic algorithm, multiactor DQN with single type of agent	Dynamic	No	Original DQN training	Energy consumption and system utility	*O*(*N*)
[9]	MINLP, DDPG with single type of agent	Dynamic	No	Original DQN training	System energy consumption	*O*(*TH*)
[29]	Q-learning with single type of agent	Static	No	Original Q-learning training with device-to-device communication	Energy consumption and latency	*O*(*TH*)
[26]	MINLP, DDQN with single type of agent	Dynamic	No	Original DDQN training	System energy consumption	*O*(*TH*)
[27]	Collaboration of FL, blockchain, and DDQL with single type agent	Dynamic	Yes	Decentralized FL training	System latency	*O*(*TH*)
Ours	Collaboration of FL and two types of DDQN agent and D3QN agent	Dynamic	Yes	Decentralized FL training with semi-global aggregation	System latency	*O*(*TH*)

**Table 2 sensors-23-02243-t002:** Main used notations.

Symbols	Definition
$h_{n,m}\left(t\right)$	$\mathrm{Wireless}\;\mathrm{channel}\;\mathrm{gain}\;\mathrm{between}\;\mathrm{mobile}\;\mathrm{device}\;u_{n}$$\;\mathrm{and}\;\mathrm{MEC}\;\mathrm{server}\;e_{m}$ in the tth time slot
$v_{n,m}\left(t\right)$	Transmission rate $v_{n,m}\left(t\right)$ between $u_{n}$ and $e_{m}$ in the *t*th time slot
$d_{n}\left(t\right)$	Computing task$\;\mathrm{generated}\;\mathrm{by}\;u_{n}$ at the beginning of the tth time slot
$s_{n}\left(t\right)$	$\mathrm{Data}\;\mathrm{size}\;\mathrm{computed}\;\mathrm{by}\;d_{n}\left(t\right)$
$c_{n}\left(t\right)$	$\mathrm{Number}\;\mathrm{of}\;\mathrm{CPU}\;\mathrm{cycles}\;\mathrm{required}\;\mathrm{to}\;\mathrm{process}\;1\;\mathrm{bit}\;\mathrm{data}\;\mathrm{of}\;d_{n}\left(t\right)$
$q_{n}^{U}\left(t\right)$	$\mathrm{Task}\;\mathrm{queue}\;\mathrm{of}\;u_{n}$
$b_{n}^{U}\left(t\right)$	Number of tasks in $q_{n}^{U}\left(t\right)$
$c_{n}^{U}\left(t\right)$	Total number of CPU cycles required to compute tasks of $q_{n}^{U}\left(t\right)$
$q_{m}^{E}\left(t\right)$	$\mathrm{Task}\;\mathrm{queue}\;\mathrm{of}\;e_{m}$
$b_{m}^{E}\left(t\right)$	Number of tasks in $q_{m}^{E}\left(t\right)$
$c_{m}^{E}\left(t\right)$	Total number of CPU cycles required to compute tasks of $q_{m}^{E}\left(t\right)$
$l_{n}^{L}\left(t\right)$	$\mathrm{Execution}\;\mathrm{delay}\;\mathrm{of}\;d_{n}\left(t\right)$ on $u_{n}$
$l_{n}^{E}\left(t\right)$	$\mathrm{Execution}\;\mathrm{delay}\;\mathrm{of}\;d_{n}\left(t\right)$ on $e_{m}$
$l_{n}^{E^{\prime }}\left(t\right)$	$\mathrm{Execution}\;\mathrm{delay}\;\mathrm{of}\;d_{n}\left(t\right)$ on a nearby MEC server $e_{m’}$
$l_{n}^{C}\left(t\right)$	$\mathrm{Execution}\;\mathrm{delay}\;\mathrm{of}\;d_{n}\left(t\right)$ on central cloud
$s_{n}^{U}\left(t\right)$	State of $u_{n}$ in the *t*th time slot
$a_{n}^{U}\left(t\right)$	Action of $u_{n}$ in the *t*th time slot
$r^{U}\left(s_{n}^{U}\left(t\right),a_{n}^{U}\left(t\right)\right)$	$\mathrm{Reward}\;\mathrm{of}\;u_{n}$ to take$a_{n}^{U}\left(t\right)$$\;\mathrm{under}\;s_{n}^{U}\left(t\right)$
$s_{n,m}^{E}\left(t\right)$	State of $e_{m}$$\;\mathrm{for}\;d_{n}\left(t\right)$ in the *t*th time slot
$a_{n,m}^{E}\left(t\right)$	Action of $e_{m}$ $\mathrm{for}\;d_{n}\left(t\right)$ in the *t*th time slot
$r^{E}\left(s_{n,m}^{E}\left(t\right),a_{n,m}^{E}\left(t\right)\right)$	$\mathrm{Reward}\;\mathrm{of}\;e_{m}$ to take$a_{n,m}^{E}\left(t\right)$$\;\mathrm{under}\;s_{n,m}^{E}\left(t\right)$
$time_{n,m}\left(t\right)$	Subsequent execution time of $d_{n}\left(t\right)$ on $e_{m}$

**Table 3 sensors-23-02243-t003:** All model parameters and simulation values.

Models	Parameters	Simulation Values
Wireless communication model	*A_d_*	24.5 dB
	*f_c_*	2595 MHz
	*D*	200 m
	*c* _0_	3 × 10^8^ m/s
	*β_n_*_,_*_m_*(*t*)	Rayleigh(1)
	*d_e_*	2.8
	*B*	20 MHz
	*p* _0_	23 dBm
	*N* _0_	−90 dBm
Wired communication model	$v_{m,m’}^{E}$	*U*(1250, 1350) Mbits/s
	$v_{m}^{E,C}$	*U*(15, 25) Mbits/s
Computing model	*s_n_*	*p*(0.3) Mbits
	*c_n_*	*U*(4, 15) cycles/bit
	*f_U_*	450 MHz
	*f_E_*	1500 MHz
Mobile device agent	*α_U_*	0.0004
	*γ_U_*	0.9
	*ε_U_*	0.9
	*CAPACITY_U_*	2000
	*BATCH_SIZE_U_*	64
	*C_U_*	100
MEC server agent	*α_E_*	0.0003
	*γ_E_*	0.9
	*ε_E_*	0.9
	*CAPACITY_E_*	1000
	*BATCH_SIZE_E_*	32
	*C_E_*	100

**Table 4 sensors-23-02243-t004:** Average task execution delay compared with baseline strategies (×10^−3^ ms).

Strategies	Single Episode	Multiple Episodes
DRT	5.125	4.968
MDL	8.468	8.445
MDO	5.889	5.280
MDR	6.056	6.000
MSL	5.510	5.113
MDC	8.064	7.271
MSR	6.833	6.550

**Table 5 sensors-23-02243-t005:** Average task execution delay compared with SMDA and OMSA (×10^−3^ ms).

Strategies	Single Episode	Multiple Episodes
DRT	5.125	4.968
SMDA	5.549	5.401
OSMA	10.992	9.993

## Data Availability

The data presented in this study are available in Table 3, Table 4 and Table 5 and Figure 5, Figure 6, Figure 7, Figure 8 and Figure 9.

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
