# Peer review of "A Federated Learning and Deep Reinforcement Learning-Based Method with Two Types of Agents for Computation Offload"

_sensors, 2023, doi:10.3390/s23042243_

Round 1

Reviewer 1 Report

In the edge-to-cloud continuum, computation offloading is a major challenge. This paper addresses it by using federated learning and deep reinforcement learning. This paper addresses the challenge of minimizing task execution delay for a multi-base station multi-mobile device MEC network that is cloud-based and collaborative. Since the problem is novel and contributions appear to be worthwhile, there are some questions about this work that remain. The following comments may be helpful in adding a few more points to this paper, filling in the gaps.

  1. There are no references for 2022 or 2023. A number of advancements have been published on this research gap, but they have not been included in the literature. It is recommended to consider the most recent works. E.g., 'Cooperative Transmission Scheduling and Computation Offloading with Collaboration of Fog and Cloud for Industrial IoT Applications'

  2. The authors are requested to compare the proposed work with recently published works.

  3. There are several performance metrics in the literature, but only a few are considered in this paper. It is recommended to consider more metrics and different scenarios to evaluate the metrics and show the superiority of the proposed work.

  4. It is necessary to provide the reasons behind the superior performance of the proposed work over the existing approaches. Provide suitable limitations as well.

  5. Deep reinforcement learning algorithms are computationally intensive. Can resource constrained devices handle such complexity on the Edge-to-Cloud continuum? It is recommended to derive the computational complexity of the algorithms presented in this paper and compare the complexity of existing approaches.

  6. DQN is explained well in this paper but mapping the problem to DQN solutions is not addressed. Can the authors clearly explain which of the computational offloading metrics corresponds to the different terminology of DQN? The authors need to provide these details clearly to avoid ambiguity.

  7. There is very little discussion of federated learning in the paper. Can the authors describe the need for federated learning in this work?

  8. Did the authors use any existing datasets to train DQN? How was the DQN trained?

  9. What was the author's method for utilizing DQN'Experiencece reply buffer in the proposal? In what ways does it benefit computational offloading decisions?

  10. It is necessary to summarize all the limitations identified in existing works. In order to make the reading experience more convenient, authors can provide a table that compares the advantages and limitations of existing works. The limitations to which the proposed work addresses are also mentioned.

  11. During the experiments, what implementation challenges were encountered, which might affect the acceptance of real-world applications?

  12. Which real-world applications are more suitable for this work?

Reviewer 2 Report

The authors propose that with the rise of latency-sensitive and compute-intensive applications, traditional cloud computing has struggled to meet the low-latency demands of applications. Mobile Edge Computing is expected to solve this problem. However, the uncertainty of computing tasks and the time-varying nature of wireless network channels make it difficult for mobile devices to make efficient computation offloading decisions. The authors present to address this problem: A federated deep reinforcement learning-based method ( denoted FDRT) with two-agent for computation offload.

This work is a contribution to the area of knowledge, but it presents some shortcomings to improve, such as:

1. - The abstract must be self-contained: "...By designing the DDQN-based mobile device agent and the D3QN-based MEC server agent..."  

2.- Several equations are presented in section 3, System Model, and section 4, Methodology. If the authors created them, proofs of correctness are missing; otherwise, show the references.

3.- Add a Discussion section regarding the results obtained. And How the problem is to be solved is reflected in the results: "...latency-sensitive and compute-intensive applications..."

Best regards

Reviewer 3 Report

The topic is closely related to the journal. Mobile edge computing paradigm enhance low sensitive and compute intensive applications compared to the cloud computing. However, there is an uncertainty for time variant wireless networks for efficient offloading computations and the researchers are targeted to solve this problem. The authors of this research present the federated deep reinforcement learning. This approach uses two-agent computation offloading. The authors also designed DDQN and D3QN mobile edge computing agents. The simulation results shows that 50% average delay reduction and optimizes the training time by 61.7%. However, efforts for better efficiency are required in the future on real-world datasets. The authors had organized the set up the Mobile Edge Computing model and explained the ne necessary components of the system including network, communication, tasks and computation with equations and mathematical expressions. The methodology and the problem statement are explained well with appropriate mathematical equations. In the federated learning system workflow for multi-agent training, authors need to explicitly be mentioned the local data set attributes. What is the difference between the “Update Local Parameter” and “upload local parameter.”

Required minor improvements:

1.     All the acronyms need the abbreviations in their first occurrence. Check this throughout the article. (Ex: DDQN, D3QN). Create a table by itself with all the acronyms and abbreviations would be a suggestion.

2.      The methodology and the problem statement are explained well with appropriate mathematical equations.

The methodology section runs almost 5 pages. A block diagram representing the sequential steps of the experiments will be helpful to the reader.

3.     Page 16, thank you for having the table with parameters and simulation values for each model. However, the table caption and number are missing.

4.     It is important to indicate the limitations of the study. It will be helpful for the audience to replicate the simulation.

5.     Future research directions are also missing.

Reviewer 4 Report

1. The abstract should be revised, it should contain bits of the following information: introduction, problems statement, aim/objectives, methodology, findings, significance of findings and a concluding statement in that order.

2.  The introduction needs to be concise and precise. Highlight the contributions.

3. List in a table all the taxonomy used in this manuscript.

4. Explain equation (12) in details.

5. Results should be present in a table form. and compare with the existing solutions.

6. Discuss results in a separate section before conclusion.

7. References section needs to revise and add some latest references from 2022/23.

Round 2

Reviewer 1 Report

None

Author Response

Dear reviewer,

Thank you.

Please see the latest version of the manuscript.

Reviewer 2 Report

The current version presents improvement, but there are three important point to improved. 

1) The propose of solución to the problem.

2) The presentation of results that support the improvements of the solution proposed. 

3) The order of the sections needs to be improved. Lenght of each section and references in equation. 

Best regards

Reviewer 4 Report

The authors of this manuscript have revised this manuscript according to my previous comments. I agree to accept this manuscript in its current form.

Author Response

(The authors gave the same response as above.)
